# Degradation of Proteins and Starch by Combined Immobilization of Protease, α-Amylase and β-Galactosidase on a Single Electrospun Nanofibrous Membrane

**DOI:** 10.3390/molecules24030508

**Published:** 2019-01-31

**Authors:** William J. Cloete, Stefan Hayward, Pieter Swart, Bert Klumperman

**Affiliations:** 1Department of Chemistry and Polymer Science, Stellenbosch University, Private Bag X1, Matieland 7602, South Africa; william.cloete@gmail.com; 2Department of Biochemistry, Stellenbosch University, Private Bag X1, Matieland 7602, South Africa; stefan.hayward@innovativeresearch.co.za (S.H.); pswart@sun.ac.za (P.S.)

**Keywords:** poly(styrene-co-maleic anhydride), enzyme, immobilization, protease

## Abstract

Two commercially available enzymes, Dextrozyme (α-amylase) and Esperase (protease), were covalently immobilized on non-woven electrospun poly(styrene-*co*-maleic anhydride) nanofiber mats with partial retention of their catalytic activity. Immobilization was achieved for the enzymes on their own as well as in different combinations with an additional enzyme, β-galactosidase, on the same non-woven nanofiber mat. This experiment yielded a universal method for immobilizing different combinations of enzymes with nanofibrous mats containing maleic anhydride (MAnh) residues in the polymer backbone.

## 1. Introduction

Enzymes play a key role in catalyzing biological reactions. As a result, they are often active under relatively mild conditions and their catalysis can be highly specific. Due to these characteristics, enzymes are increasingly being used in modern biotechnological settings, such as the food, biofuel and fine chemical industries [1]. However, enzyme purification from its native source is often difficult and costly [2]. Due to the increased demand for industrial enzymes, a sizable portion of commercial enzymes are, therefore, recombinantly produced [3,4]. Advances in recombinant enzyme technology have enabled the production of large quantities of robust enzymes, specifically functionalized to meet industrial demands [5,6]. Presently, commercial enzymes are available both freeze-dried and in solution. However, advances have also been made in immobilization of enzymes for use in batch reactors [7]. Enzymes may be immobilized by adsorption or covalent attachment to polymers and other high-surface-area substrates that find application in fine-chemical synthesis, fabrication of biosensors, food processing, protein digestion, and bioremediation [8].

Immobilized enzymes have the advantage over free enzymes in solutions in that they can be used repeatedly, and the risk of product contamination is significantly reduced. However, in certain cases, the immobilization substrate matrix (immobilization support) may influence the catalytic activity of the immobilized enzymes, depending on the characteristics of the substrate [9]. Enzyme immobilization on mesoporous ceramics, for instance, can lead to the loss of catalytic activity because the enzyme is contained within the support. This containment prevents diffusion of the enzyme substrate to the active site and release of the product from the enzyme active site [8,10]. Nonwoven nanofibrous mats can overcome this disadvantage because their high level of porosity and interconnectivity allow mass transfer of the substrates and products. The high surface to volume ratio of nanofibrous mats, furthermore, provides a relatively large surface area for enzyme immobilization, which results in higher enzyme loadings compared to other solid supports [11]. Nanofibrous mats could also be spun from a variety of polymers allowing for site-directed immobilization of enzymes (e.g., via a polyhistidine tag) [12]. This would enable the design and manufacture of tailor-made surfaces that limit the loss of enzymatic activity.

In a study by Li et al. [13], the use of enzymes immobilized on high surface area nanofibrous materials for the conversion of raw materials into value added products is described. In this study, the authors hydrolyzed soybean seed oils using a lipase from Candida rugosa immobilized on polyacrylonitrile (PAN) fibers, illustrating the feasibility of this technology. Enzyme immobilization also allows for greater control of the product since contact time can be optimized to suit specific demands. For this reason, immobilized enzymes are preferential to batch solution processes, which allow very little control over the product. The immobilization of enzymes on high-surface-area fiber mats could also potentially increase their economic feasibility for industrial use. For instance, a major drawback of converting starch into maltose using α-amylase enzymes is that this conversion is performed in a batch reaction, limiting the possibility to recover the enzyme after use [14]. In an industrial setting, this would result in increased production costs. Immobilized enzyme technology allows for reuse of the enzyme, which is one of the key expenses and most vital components of the process.

Furthermore, enzymes are not only widely used in processes to convert bio-matter into value added products, but also in the remediation of biofouling that occurs in drainage systems, filtration processes, and bioreactors [15]. Biofouling is a common phenomenon in moist nutrient-rich environments, such as the drainage systems of abattoirs and dairies [16]. In addition to biofouling, the formation of biofilms could also pose a health risk. Mature biofilms can lead to the proliferation and spreading of pathogenic bacteria, which would necessitate frequent cleaning and strict sterilization regimes, resulting in additional costs. Although chemical treatments may be effective if performed correctly, the chemicals used are often harmful to the environment. It has been shown that enzymes such as proteases and amylases could be used for the remediation of biofilms, since biofouling can be drastically decreased in the presence of these enzymes [17]. In a quest to create anti-biofouling coatings and surfaces, Cordeiro et al. [18] showed that immobilized enzymes are able to counteract biofouling, and subsequently concluded that co-immobilization of different enzymes is the most effective route to inherently anti-biofouling surfaces. Therefore, a definite need exists for the capability to immobilize different enzymes onto the same matrix.

Currently, the best strategy to prevent the formation of biofilms is to inhibit initial adhesion of microorganisms. However, the size, shape, and nature of the biofilms, and the extracellular polymeric substances (EPS), are not yet fully understood. Due to this complexity, immobilization of multiple enzymes on the same surface is often required to prevent the formation of mature biofilms. Application of combinations of enzymes have shown promise in so-called clean-in-place systems related to the food industry. However, since each biofilm differs with relation to the microbial community that produces it, different combinations of immobilized enzymes need to be screened to establish the optimal combination of immobilized enzymes for each specific application.

It is known that maleic anhydride (MAnh) residues spun into nanofibers can be used as a solid support for the immobilization of enzymes [18,19,20,21,22,23]. Our research group has developed a facile approach to enzyme immobilization on electrospun nanofibers containing MAnh residues. This approach does not require rigorous modification or activation steps prior to immobilization, and therefore may provide a low-cost alternative for industrial applications [24]. Here, we extend the previous work. We report on the immobilization of commercial proteases and α-amylases on electrospun MAnh-containing nanofiber mats to pioneer the manufacturing of membranes that are inherently anti-biofouling or could be used in membrane reactors for application in the food industry. The primary role of proteases is the hydrolysis of proteins, resulting in the liberation of smaller peptides and amino acids. α-Amylases, on the other hand, hydrolyze starch into mono- and oligosaccharides. In this manuscript, we assess the ability of two enzymes, a commercial protease (Esperase) and α-amylase (Dextrozyme), immobilized on electrospun nanofibrous materials, to degrade proteins and starch in solution, both separately and in combination. The ability to immobilize multiple enzymes is further extended through the inclusion of an additional enzyme, β-galactosidase (β-gal), in combination with one or both of the other enzymes used in this study. In doing so, we sought to prove that employing nanofibrous mats, containing reactive MAnh residues in the polymer back bone, can be used as a universal method for immobilizing a variety of enzymes with retention of their catalytic activity.

## 2. Materials and Methods

### 2.1. Reagents

All reagents were used as received, and all buffer solutions were previously prepared and used as stock solutions and diluted as needed. Unless stated otherwise, all reagents and chemicals used were of analytical grade. Chemical grade azo-casein, used for the protease activity assay, and the Ceralpha α-amylase assay kit were purchased from Megazyme, Wicklow, Ireland. The o-nitrophenol-β-d-galoctoside (ONPG) used for determination of β-gal activity was 99.0% pure and was obtained from Sigma-Aldrich, Schnelldorf, Germany. All buffer reagents, polystyrene cuvettes, 96-well microtiter plates (Greiner), phenylmethane sulfonyl fluoride (PMSF), and the Pierce BCA protein determination kit were also obtained from Sigma-Aldrich.

### 2.2. Enzyme Assays

Commercially available enzyme kits and previously described methods were used to assess the enzyme activities of both the free and immobilized enzymes. Protease activity was determined using azo-casein as a substrate, as previously described by Sheng-Feng Li et al. [13]. Amylase activity was determined using the Ceralpha α-amylase assay procedure (AOAC Method 2002.01, Megazyme International, Wicklow, Ireland). The activity of immobilized β-gal was determined using the method described by P. Held using ONPG (Sigma Aldrich) as a substrate [25]. All the assays were chosen so that each substrate, upon conversion, would lead to the formation of a readily detectable UV-active dye. The absorbance for the protease, and α-amylase assays were determined using a Cary 60 UV-Vis spectrophotometer (Agilent Technologies, Santa Clara, CA, USA). β-gal activity was discontinuously monitored using a BioTek PowerWave microtiter plate reader (BioTek Instruments, Winooski, VT, USA). The determined absorbance was subsequently used to quantitatively determine and compare the enzyme activity of both the free and immobilized enzymes. In all cases, bovine serum albumin, immobilized on an electrospun nanofibrous mat, was used as a control. Enzyme loading was decided by determining the protein concentration of the stock enzyme solutions prior to, and after, immobilization. The protein content of each wash step was also determined.

## 3. Experimental Section

### 3.1. Electospinning of Poly(styrene-alt-MAnh) Nanofibrous Mats

The synthesis of poly(styrene-*alt*-maleic anhydride) and subsequent electrospinning into nanofibres were performed as previously described by Cloete et al. [24] Briefly, the polymer was produced by copolymerizing, in solution, a 1:1 molar ratio of styrene and maleic anhydride under inert conditions using AIBN as an initiator at 60 °C. Electrospinning of the polymer was done from a 1:2 DMF/acetone solution (15 wt%) with the collector at a distance of 15 cm and offset of 4 cm from the needle tip that was connected to a high voltage supply (25 kV, 400 μA, 10 W) set at 15 kV, and the polymer solution was fed at 0.01 mL·min^−1^ by means of a feed pump (Harvard, Model 33 Twin Syringe Pump, Holliston, MA, USA).

### 3.2. Enzyme Immobilization

Protein immobilization was achieved by incubating a 4 cm^2^ fiber mat in 5 mL protein solution (ca. 43 mg/mL for protease and ca. 296 mg/mL for α-amylase, used as supplied without dilution) for 1 h at room temperature with gentle agitation. As a control, bovine serum albumin (10 mg/mL) was also concurrently immobilized on a separate mat and in combination with both protease and α-amylase. When co-immobilization with the protease was performed, the protease activity was reversibly inhibited using phenylmethane sulfonyl fluoride (PMSF) to prevent loss of activity via protein hydrolysis. Bovine serum albumin was used as an immobilization control to verify if immobilization of protease and α-amylase can be achieved in combination with an arbitrary additional protein. This was necessary because the enzyme solutions in this study were commercially available and used as supplied, without purification and reconstitution prior to immobilization. Samples of each protein solution (1 mL) were collected prior to incubation, and the protein concentration was subsequently determined as described below. After incubation, each fiber mat was extensively washed with phosphate buffered saline (PBS, pH 7.0) containing 0.1% Tween 20 (4 × 5 min) to remove non-covalently bound protein. Four 1 mL aliquots of the PBS-Tween wash solutions were collected, and the protein content was determined, together with the original protein solution collected prior to incubation with the membrane, using the Pierce BCA protein assay kit with bovine serum albumin as standard. The amount of immobilized protein was calculated as the difference in protein content prior to- and following immobilization. The same immobilization procedure described above was used for all enzyme and control solutions and is graphically depicted in Figure 1.

### 3.3. Enzyme Activity Assays

#### 3.3.1. Protease Activity Assay

During protease immobilization studies, the protease was evaluated for retention of enzymatic activity using an assay adapted from Li et al. [13]. Azocasein substrate solution (2.5 mL of 2.5% stock in 50 mM borax buffer, pH 9.5) was added to the electrospun fibre mat containing 0.6 mg immobilized protein. The reaction was quenched with 2.5 mL of 10% trichloroacetic acid (TCA) in deionized water, after 5 min incubation at 30 °C. The solutions were held at a constant pH of 9.5 to simulate the conditions under which Esperase is used in laundry detergents. After centrifugation of each reaction mixture, the UV absorbance of the supernatant was determined at 340 nm. The rate at which the immobilized protein hydrolyzed the azocasein substrate was calculated using Equation (1), where Δ*A* represents the change in absorbance at 340 nm, V the reaction volume (in mL), *ε* is the extinction coefficient of the product of azocasein hydrolysis at 340 nm (38 mM^−1^·cm^−1^) (Li et al. [13]), *l* is the optical light path length of 1 cm, and t is the reaction time of 5 min.

#### 3.3.2. Amylase Activity Assay

The Ceralpha method (AOAC Method 2002.01) was used to quantify α-amylase activity of the immobilized and free enzyme using the Megazyme α-amylase kit (Megazyme International) as per manufacturer instructions. Briefly, 10 mL of the substrate solution, ONPG, was equilibrated to 40 °C separate from the fiber mat samples. After 5 min, 800 µL of the substrate was added to each fiber mat and incubated for exactly 10 min at 40 °C. After incubation, enzyme activity was quenched by the addition of 8 mL 1% Tris-base. A sample blank was prepared by incubating 200 µL liquid enzyme solution with 8 mL 1% Tris-base prior to the addition of the substrate. Incubation was performed as described with the fiber mat samples. The samples were then thoroughly mixed, and the absorbance of the supernatant was determined at 400 nm against the sample blank. Total enzyme activity was calculated using Equation (1), where ∆*A* represents the change in absorbance at 400 nm following incubation (final absorbance—blank absorbance), *V* is the total reaction volume in mL, *ε* is the millimolar extinction coefficient of para-nitrophenol (18.1 mM^−1^·cm^−1^, as per kit instructions), *l* is the optical light path length of 1 cm, and *t* is the total incubation time of 10 min for a surface immobilized area of 4 cm^2^.
(1)$$U/mg\;protein/cm^{2}=\frac{\left(∆A\times V\right)}{\left(t\times \varepsilon \times l\times \left(\frac{\left[protein\right]}{cm^{2}}\right)\right)}$$

#### 3.3.3. Co-Immobilization of Multiple Enzymes on the Same Surface

Co-immobilizations of Dextrozyme and Esperase, inhibited with PMSF (Thermo Scientific, Rockford, IL, USA), were mixed in a buffered solution and added to a 4 cm^2^ membrane, as with the immobilization of individual enzymes. PMSF reversibly inhibited the catalytic activity of the protease, preventing the degradation of the α-amylase during immobilization. Subsequent to immobilization, the PMSF was removed along with any non-covalently bound enzyme during the wash steps, with PBS Tween-20 to restore protease activity. The enzymatic activity of Dextrozyme was assayed as described previously and calculated in the same way as before, using Equation (1).

A co-immobilization experiment with Dextrozyme, Esperase, and commercial β-gal was also performed using nanofibrous mats obtained through a high-throughput industrial electrospinning process using commercially available poly(styrene-*co*-MAnh) (SMA) [XIRAN SZ28110, Polyscope, 28 wt% MAnh, M_w_ = 110 kDa, *Ð* ~ 2]. SMA was provided to the Stellenbosch Nanofiber Company (Cape Town, SA) who optimized the electrospinning conditions for large scale production, and the electrospun nanofibrous mats were used as received. Protease and α-amylase activities were determined as described above. The activity of immobilized β-gal was determined using OPNG as substrate. A discontinuous assay was used to determine enzyme activity alongside a fiber mat containing immobilized BSA as the assay blank. The substrate solution consisted of 100 mM sodium phosphate buffer (pH 7.0), 0.1 mM MgCl_2_, 50 mM β-mercaptoethanol, and 1.33 mg/mL ONPG. The final solution was equilibrated at 37 °C in a water bath. Of the equilibrated substrate solution, 5 mL was added to 4 cm^2^ mats with immobilized β-gal. 100 µL aliquots were collected at 10 s intervals for the β-gal and amylase—β-gal co-immobilized samples and at 1-min intervals for the α-amylase—β-gal-protease co-immobilized samples. Each aliquot was added to a 100 µL 2% Tris base in the wells of a 96 well plate. The UV absorbance of the solution in each well was subsequently determined at 420 nm with a BioTek PowerWave HT plate reader (Biotek, Broadview, IL, USA).

## 4. Results and Discussion

The proposed mechanism for enzyme immobilization in the current approach works through a nucleophilic reaction of the primary amines of lysine residues with MAnh residues of the SMA copolymer. Since the 3D structures of enzymes are not rigid, all available lysine residues are equally likely to undergo the amidation reaction to the reactive MAnh residues. This probability could result in a random orientation of the immobilized enzyme, limiting substrate access to the active site [26,27]. For this reason, enzyme immobilization is often associated with a decrease in catalytic activity when compared to native enzymes in solution [26,28].

Based on the results presented in Table 1, both the protease and α-amylase were successfully immobilized on the electrospun nanofibrous mat individually and in combination. Both enzymes partially retained their catalytic activity; however, immobilization did result in an overall decrease in enzyme activity. Interestingly, when immobilized in combination with the protease, α-amylase activity showed a 3-fold increase compared to individually immobilized α-amylase. On the other hand, the activity of the protease decreased by ca. 98% when co-immobilized with α-amylase. Similar results have been reported for co-immobilized cholesterol oxidase (COD) and horseradish peroxidase (HRP) [29]. In this case COD and HRP make use of the same substrate, H_2_O_2_, and the increase in activity was proposed to be due to enhanced mass transfer effects and the proximity of the immobilized enzymes [29]. This phenomenon was also proposed to be the mechanism for the observed increase in activity of co-immobilized glucoamylase and glucoisomerase, which similarly make use of the same substrate [30]. Regarding changes in activity for enzymes employing different substrates, an increase/decrease in activity cannot be readily explained through mass transfer and proximity alone.

Since amylase activity was increased when co-immobilized with a protease, it is possible that α-amylase activity was augmented by proteolytic activation. It has been reported that some enzymes, especially those involved in host defenses, are post-translationally activated by proteolysis [31]. The marked reduction in protease activity upon co-immobilization with α-amylase could be because of immobilization efficiency. As discussed earlier, enzyme immobilization relies on the availability of free lysine residues. Since the primary structures of the protease and α-amylase used in this study are not known, it is possible that the α-amylase has more available lysine residues that could undergo amidation to the reactive MAnh residues, resulting in increased α-amylase loading. However, since it was not possible to quantitatively determine the loading of individual enzymes in this study, this possibility could not be confirmed. Although post-translational activation and enzyme loading could potentially explain the results obtained, the exact mechanism for the observed increase in α-amylase activity and the concurrent decrease in protease activity could not be confirmed. This is, therefore, a possible avenue for future investigation.

To prove the commercial viability of the immobilization process, and to verify whether the co-immobilization of protease and α-amylase is not in fact a special case, the experiments were extended to include the immobilization of an additional enzyme, β-gal. As can be seen in Figure 2, β-gal activity was retained once immobilized individually and in combination with protease and α-amylase. However, as can be seen in Figure 2, a reduction in β-gal activity was observed when co-immobilized with only α-amylase. A further reduction in β-gal activity occurred when co-immobilized in combination with both α-amylase and protease.

Determination of α-amylase activity when co-immobilized with protease and β-gal indicated that enzyme activity is partially retained regardless of the combination of immobilized enzymes. As can be seen in Table 2, α-amylase activity was once again augmented when immobilized in combination with the protease. Since α-amylase activity is only augmented when the protease is also present, this result further indicates a proteolytic activation mechanism for the amylase enzyme.

## 5. Conclusions

The results presented in this study demonstrate that commercial protease and α-amylase enzymes could be immobilized on SMA nanofibrous mats with partial retention of catalytic activity. Co-immobilization of protease and α-amylase led to a marked increase in α-amylase activity with a concurrent decrease in protease activity. It was also demonstrated that α-amylase retains catalytic activity when co-immobilized with protease and an additional enzyme, β-gal, on the same nanofibrous mat. Recommended future research includes an investigation of additional combinations of co-immobilized enzymes, and the evaluation of membranes with immobilized enzymes for continuous use in the digestion of complex solutions of biomolecules, which may allow them to withstand biofouling. A combination of immobilized enzymes targeted at the digestion or degradation of a variety of proteins and other biomolecules will go a long way toward providing surfaces that prevent the attachment and proliferation of microbial foulants. This opens the door to designing single membranes that can perform cascade reactions or single membranes that can digest complex solutions of proteins and other biomolecules. Nanofibrous mats are feasible and promising materials for use as filtration membranes [32,33]. The design of tailor-made nanofibrous filter materials with immobilized enzymes may provide a solution for the prevention of biofouling by excluding organic fouling agents or converting and metabolizing nutrients that sustain biofilm growth. The ultimate application would be a means to pre-treat the feed in water purification, as in the remediation of effluents and waste streams, or for use in bioreactors, to convert starch and oils into feedstock for the food industry.

## Figures and Tables

**Figure 1 molecules-24-00508-f001:**
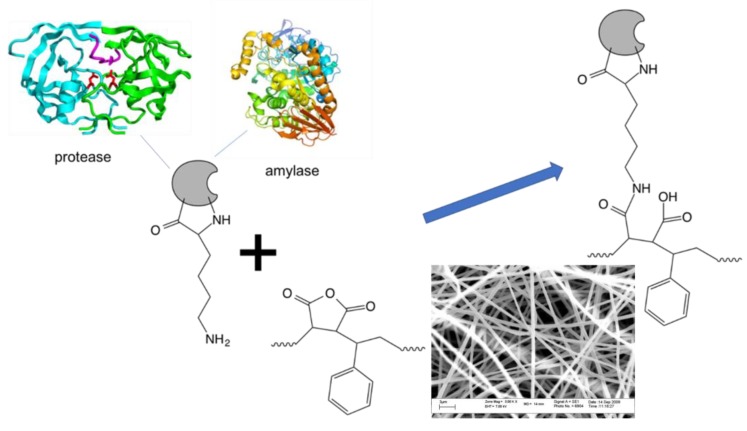
General scheme for immobilization of enzymes on poly(styrene-*alt*-maleic anhydride) nanofibers.

**Figure 2 molecules-24-00508-f002:**
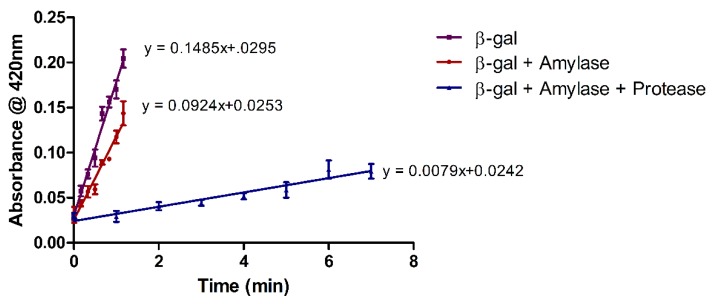
Activity progress curves for β-gal immobilized individually on an SMA nanofibrous mat, and in combination with α-amylase, as well as a combination of α-amylase and protease. Activity was spectrophotometrically determined as a function of time using ONPG as substrate. All results are shown as the mean ± *SD* of triplicate experiments (n = 3).

**Table 1 molecules-24-00508-t001:** Enzymatic activity and protein loading of enzymes immobilized on poly(styrene-*alt*-maleic anhydride) nanofibrous mats (n = 3).

Enzyme	Enzyme Loading ^a^	Activity ^b^	Free Enzyme Activity ^b^	% Retention ^c^
protease	7.54 ± 0.98	7.78 × 10^−4^ ± 1.32 × 10^−5^	8.87 × 10^−4^ ± 8.58 × 10^−6^	87.0%
α-amylase	39.7 ± 0.42	0.808 × 10^0^ ± 2.78 × 10^−3^	8.81 × 10^0^ ± 0.175 × 10^0^	9.0%
protease + α-amylase	N/A	1.54 × 10^−5^ ± 4.40 × 10^−7^	8.87 × 10^−4^ ± 8.58 × 10^−6^	1.7% (protease)
	N/A	2.37 × 10^0^ ± 1.17 × 10^−1^	8.81 × 10^0^ ± 0.175 × 10^0^	27.0 % (α-amylase)

^a^ in mg·cm^−2^. ^b^ in µmol·min·mg·cm^−2^. ^c^ is the activity retained after immobilization vs. the activity of the free enzyme.

**Table 2 molecules-24-00508-t002:** Enzymatic activity of α-amylase immobilized on its own and in combination with β-galactosidase and protease on poly(styrene-*co*-maleic anhydride) nanofibrous mats (n = 3).

Enzyme	Total Enzyme Loading (mg)	Amylase Activity (µmol·min·mg·cm^−2^)
α-amylase	35.2 ± 0.47	16 × 10^−3^ ± 2.40 × 10^−4^
α-amylase + protease	10.5 ± 3.81	36 × 10^−3^ ± 2.60 × 10^−4^
α-amylase + protease + β-galactosidase	18.3 ± 1.58	5 × 10^−3^ ± 1.11 × 10^−5^
BSA	1.35 ± 0.63	N/A

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
