# Peer review of "Degradation of Proteins and Starch by Combined Immobilization of Protease, α-Amylase and β-Galactosidase on a Single Electrospun Nanofibrous Membrane"

_molecules, 2019, doi:10.3390/molecules24030508_

Reviewer 1 Report

The manuscript entitled “Degradation of Proteins and Starch by Combined Immobilization of Protease, α-Amylase and β-Galactosidase on a Single Electrospun Nanofibrous Membrane” submitted by William J. Cloete et al. proposes a method for co-immobilizing of different enzymes on fibrous membranes prepared by electrospinning. 

The topic is interesting but due to the lack of the results and experiments I could not recomand for publication. Some comments and suggestions are presented below:

The introduction is too long and too general ( at the beginning). 

The use of b-galatosidase is not motivated. 

There is a lack of the date discussed in the text but not presented.

Were the activities of the combined native enzymes determined before immobilization? Was the addition of the protease effect on the b-gal or amylase activity considered?

A fully charaterization of the immobilized enzymes (pH and temperature stability, reusability in several reaction cycles) is needed. 

Reviewer 2 Report

The manuscript entitled “Degradation of Proteins and Starch by Combined Immobilization of Protease, α-Amylase and β-Galactosidase on a Single Electrospun Nanofibrous Membrane” submitted by William J. Cloete et al. proposes a method for co-immobilizing different combination of enzymes on fibrous membranes prepared by electrospinning. The methodology for enzyme co-immobilization was described and the enzyme activity was assessed after the co-immobilization of different enzymes.

This paper is an extension of a previous paper (Polym. Chem 2011, 2, 1479) and provides additional results on the methodology developed through the co-immobilization of three commercial enzymes with antifouling behaviour. The approach is interesting and the study could become technically sound. Nevertheless, I cannot recommend it for publication (at least in the current state). Some comments for revision:

1. One of the main results is the increase of the amylase activity and the decrease of the protease activity after the co-immobilization. The authors hypothesize that the available lysine residues that undergo amidation to the reactive MAnh residues could be related with the activity increase in amylase and that the reduction in protease activity could be related to immobilisation efficiency. However, the authors do not confirm the hypothesis, they only indicate that additional research is necessary. This hypothesis should be tested experimentally with additional experiments. In my opinion, this is a key point in this paper.

2. More information is needed about the experiments and statistics. What was the "n" (number of independent replications)? My impression is that the data reported are from one-shot experiments on single samples. If so, the study is too small to be publishable.

3. Materials and methods (pg. 3, line 102). The authors indicate that “all reagent and chemical used were of analytical grade, and was obtained from reputable chemical suppliers.” In any case, the chemical suppliers of all reagents (also included the materials to synthetize the electrospun membranes) should be included.

4. Experimental section (pg. 3, line 127). Although the author referred to a publish paper for the fibrous electrospun membranes, a brief summary of the methodology should be included to facilitate understanding of the paper. 

5. Experimental section. Pg. 5, line 189. The authors assess the methodology by a co-immobilisation experiment with three enzymes. This experiment including β-galactosidase was performed with electrospun membranes (commercial) prepared using a different methodology from the one used for protease and amylase. Please, justify the reason for doing so.

6. This additional membrane used for the Dextrozyme, Experase and β-galactosidase co-immobilization should be indicated in the Materials and methods section.

7. Please check the values for protease activity in Table 1 and line 218 (pg 250).

8. Results and discussion section. Pg. 7, line 254. The authors state: “Determination of protease, and α-amylase activities co-immobilised with β-gal indicated that enzyme activity is retained regardless of the combination of immobilised enzymes”. Please, indicate the results that sustain that statement. In Table 2, only the amylase activity is reported.

9. Conclusions. Pg. 7, line 265. The authors indicate: “It was also demonstrated that α-amylase retains catalytic activity when co-immobilised with protease and an additional enzyme, β-gal on the same nanofibrous mat.” This sentence, from my point of view, should to be rephrased. The catalytic activity of α-amylase increase when co-immobilised with protease, nevertheless, when β-galactosidase is added, the activity of this enzyme decrease significantly (i.e., the catalytic activity is not completely retained).

Some other minor points:

1. Introduction. Lines 75-82: Some references should be included in this paragraph.
2. The scale bar in the SEM picture (Figure 1) is too small. It is really difficult to see it properly.
3. There is no reference to Figure 1 in the text.
4. Please increase the font size in equation 1. Some of the symbols are too small.  

Author Response

see attached file

Round  2

Reviewer 2 Report

The authors have addressed almost everything I raised. Nevertheless, the reviewer insists on two points previously indicated:

1. I asked about experiments and statistics. In their response, the authors say:

‘All immobilization experiments were done in triplicate and the results are presented as an average of the experiments’.

Why is not this information included in the manuscript?

 2. The authors referred to a publish paper for the fibrous electrospun membranes and also provide the context in the introduction. I insist, however, that a brief summary of the methodology should be included in the experimental section. 

Author Response

We have made the requested minor changes to the manuscript.

we have included the number of repeat experiments explicitly in the caption of the relevant tables.

we have provided a brief description of the polymerization and electro spinning processes in the experimental section.